# TASK GENERALIZATION IN DECISION-FOCUSED LEARNING

## ABSTRACT

Real-world optimization problems often contain uncertain parameters that must be predicted prior to solving. For example, a delivery company must make its routing decisions when the traffic conditions, and thus the road traversal times, are uncertain. The models used to predict these uncertain quantities are commonly trained in a way that is agnostic of the optimization problem and that focuses solely on predictive accuracy. However, such a prediction-focused training procedure generally does not minimize the downstream task loss of interest (e.g., the suboptimality of the roads that are selected based on the predictions). This has led to the development of decision-focused learning (DFL) methods, which specifically train the predictive model to make predictions that lead to good decisions on the considered optimization task. However, as we show in this paper, such models often generalize poorly to altered optimization tasks. For example, in the context of a routing problem, their performance may deteriorate when the destination node changes. To improve on this, we first explore how the model can be trained to generalize implicitly, by simply training it on different tasks sampled at training time. We then propose a more sophisticated approach by adding the use of explicit learned task representations, to enable the model to adapt its predictions better to different tasks. To this end, we represent the optimization problems as bipartite variable-constraint graphs, and train graph neural networks (GNNs) to produce informative node embeddings that are then given to the predictive model. In our experiments, we start by showing that the state of the art in DFL tends to overfit to the specific task it is trained on, and generalizes poorly to changing tasks. We then show that both of our proposed approaches significantly improve on this, with the explicit task representations generally providing an additional improvement over the implicit approach.

## 1 INTRODUCTION

Decision-making processes are commonly modeled as constrained optimization problems, which are solved by optimizing an objective function, subject to certain constraints. However, in many applications, some of the problem's parameters are not known with certainty at the time at which the problem must be solved. For example, a delivery company must make its high-level routing decisions with respect to uncertain future traffic conditions. These traffic conditions must be predicted before a solution can be computed.

Such problems can be framed as *predict-then-optimize problems*, which consist of a prediction stage and an optimization stage. In the prediction stage, a machine learning model is used to predict values for the unknown parameters. In the optimization stage, the resulting optimization problem instantiated by the predictions is solved. The quality of the final solutions hinges on the predictive model used in the first stage, making the training procedure of this model highly important. Two training paradigms can be distinguished in this context.

In *prediction-focused learning* (PFL), the predictive model is trained to maximize the accuracy of the predictions, using traditional regression losses like the mean-squared error or negative log-likelihood. In other words, prediction-focused learning does not consider the downstream optimization problem at training time. Though simple and efficient, such training generally leads to suboptimal decision-making.

In *decision-focused learning* (DFL), the predictive model is trained in order to maximize the quality of the solutions obtained after the optimization stage with respect to the ground-truth optimization problem parameters. In other words, the optimization problem is integrated into the predictive model's training procedure. As numerous works have shown, models trained with such a procedure generally make predictions that lead to better decisions than models trained in a prediction-focused manner (Amos & Kolter, 2017; Donti et al., 2017; Wilder et al., 2019; Pogančić et al., 2020; Mulamba et al., 2021; Elmachtoub & Grigas, 2022; Mandi et al., 2022).

The two main challenges involved in DFL are related to backpropagation and scalability. First, gradient-based DFL is not straightforward when the optimization problem is combinatorial, since this makes the gradients of the task loss with respect to the predictive model's parameters zero almost everywhere. Second, DFL suffers from poor scalability, since it involves the repeated solving of optimization problems in the training loop of the predictive model. Most works have focused on tackling these two challenges.

In this work, we identify and address an open problem that has thus far received much less exposition: the limited ability of decision-focused models to generalize across new optimization tasks. Existing DFL methods lead to models that perform well on the optimization problem that they have been trained on, but that generalize poorly to changing tasks, i.e., changing constraints. This is a serious limitation of the state of the art, as real-world optimization problems change *all the time*. The available resources in scheduling problems, the traversable edges in graph problems and the customer demands in vehicle routing problems, are all subject to constant change.

To investigate and enable task generalization, we make the following contributions

- We formalize the task generalization problem and show that state-of-the-art DFL methods do not generalize well over new, unseen tasks. That is, they tend to overfit to the training task, leading to significantly worse decision-making on altered tasks.
- We propose two approaches for improving the task generalizability of DFL models. The first involves training on different tasks implicitly, that is, without representing the task explicitly in the input to the predictive model. The second additionally uses explicit learned task representations, through node embeddings produced by a graph neural network (GNN) that is applied to the bipartite variable-constraint graph associated with the task.
- We extensively evaluate our proposed approaches on predict-then-optimize problems in which the task is altered at test time in various ways. Our findings show that both approaches significantly improve the model's ability to adapt to changing tasks, and that on some task generalizations, the explicit task representations offer a considerable additional improvement over the implicit approach.

## 2 DECISION-FOCUSED LEARNING

The DFL paradigm aims to solve predict-then-optimize problems. These problems involve the prediction of unknown parameters $c$ of a parametric constrained optimization problem. Here, we specifically focus on linear programming (LP) and integer linear programming (ILP) problems, which have the following form (where equation 1d is only present for ILPs):

$$z^\star(c) = \arg\min c^\top z \tag{1a}$$
$$\text{s.t.} \quad Az \leq b \tag{1b}$$
$$z \geq 0 \tag{1c}$$
$$z \in \mathbb{Z}^n \tag{1d}$$

For any given value of $c$, solving the optimization problem involves finding a solution $z^\star(c) \in \mathbb{R}^n$ that optimizes objective function $c^\top z$ while being feasible with respect to the constraints.

Like most DFL works, we assume that constraint parameters $A$ and $b$ in (1) are known, and objective parameters $c$ are unknown, but correlated with known features $x$ according to some joint distribution. Given is a set of instances $D = \{(x_i, c_i)\}_{i=1}^N$ coming from this distribution. This dataset is used to train a predictive model $m_\omega$ – usually a neural network – which makes predictions $\hat{c} = m_\omega(x)$.

In predict-then-optimize problems, the goal in training is *not* to optimize the predictive accuracy of $\hat{c}$ with respect to ground-truth $c$. Rather, one aims to make predictions $\hat{c}$ that lead to 'good'

decisions $z^\star(\hat{c})$ with respect to ground-truth parameters $c$. In other words, the goal is to learn to make predictions $\hat{c}$ that perform well with respect to some task loss $\mathcal{L}(z^\star(\hat{c}), c)$. The aim is to minimize the *expected task loss*:

$$\min_\omega \mathbb{E}_{x,c\sim P}[\mathcal{L}(z^\star(m_\omega(x)), c)] \tag{2}$$

A common task loss, and the one we will use in the experimental evaluation of this work, is the *relative regret*. The relative regret captures the suboptimality of the made decisions $z^\star(\hat{c})$ with respect to the ground-truth parameters $c$. For linear objectives, the relative regret is computed as:

$$Regret(z^\star(\hat{c}), c) = \frac{c^\top z^\star(\hat{c}) - c^\top z^\star(c)}{c^\top z^\star(c)} \tag{3}$$

While training a model to minimize the task loss is desirable, it is not straightforward when the optimization problem is combinatorial. This is because for such problems, the gradients of the task loss are zero almost everywhere. This can be seen when applying the chain rule:

$$\frac{\partial \mathcal{L}(z^\star(\hat{c}), c)}{\partial \omega} = \frac{\partial \mathcal{L}(z^\star(\hat{c}), c)}{\partial z^\star(\hat{c})} \frac{\partial z^\star(\hat{c})}{\partial \hat{c}} \frac{\partial \hat{c}}{\partial \omega} \tag{4}$$

The second factor $\frac{\partial z^\star(\hat{c})}{\partial \hat{c}}$ expresses the change in $z^\star(\hat{c})$ when $\hat{c}$ changes infinitesimally. However, because of the combinatorial nature of the optimization problem, this factor is zero almost everywhere, and non-existent where it is not zero. In other words, a small change in the problem's parameters $\hat{c}$ either does not change its solution $z^\star(\hat{c})$, or changes it discontinuously (leading to non-existent gradients). Most existing work has focused on circumventing this obstacle. This can be done by analytically smoothing the optimization problem (Wilder et al., 2019; Mandi & Guns, 2020), by smoothing through perturbation (Pogančić et al., 2020; Berthet et al., 2020; Niepert et al., 2021), or by the use of a surrogate loss function that is smooth but still reflects the decision quality (Elmachtoub & Grigas, 2022; Mulamba et al., 2021; Mandi et al., 2022). For a comprehensive overview of existing methods, we refer the reader to Mandi et al. (2023).

These methods have in common that they tailor the predictive model to *a single optimization task*, i.e., a single set of constraints. One exception is (Tang & Khalil, 2022b), in which the model is trained for multiple tasks at once. They consider a setting in which there are $T$ tasks, which are all fully known, and which are used both to train the model and to evaluate it at test time. Each task $t$ corresponds to a feasible space $(A_t, b_t)$ (as in (1)), and a corresponding task-specific loss $\mathcal{L}_t$. The model is trained on an aggregate of the task-specific losses, e.g., $L = \sum_{t=1}^{T} \mathcal{L}_t(z_t^\star(m_\omega(x)), c)$, where $z_t^\star(m_\omega(x))$ is the optimal solution for cost vector $m_\omega(x)$ for task $t$.

While Tang & Khalil (2022b) consider a multi-task setting, they do not consider *task generalization*. That is, the ability of the model to generalize across new tasks, which have not been seen at training time. To the best of our knowledge, no existing work has considered this aspect. Yet this is highly important, as real-world optimization problems are not static, and contain constraints that can be expected to change in ways that cannot be fully enumerated at training time.

## 3 TASK GENERALIZATION

When we speak of task generalization, we do not consider a fixed set of $T$ tasks (as in the multi-task setting considered in (Tang & Khalil, 2022b)), but instead consider a distribution over tasks $\mathcal{T}$. When training a model to generalize over tasks, the goal in training changes from the minimization of the task loss on a single task (equation 2) into the minimization of the *expected task generalization loss*. We formalize this as a double expectation:

$$\min_\omega \mathbb{E}_{t\sim\mathcal{T}} \mathbb{E}_{x,c\sim P}[\mathcal{L}_t(z_t^\star(m_\omega(x)), c)] \tag{5}$$

To achieve good task generalization performance, we consider two approaches. The first approach (Section 3.1) is to train the model *implicitly* (i.e., without representing the task in the input to the model) – in a decision-focused manner – on many different tasks $t$, sampled from $\mathcal{T}$. This has the advantage that the model will be less prone to overfit on any specific task. However, a potential disadvantage is that the model will also lose its ability to tailor its predictions specifically to any given task. It will learn to make predictions that, on average, work well across $\mathcal{T}$, but that don't necessarily excel on the specific tasks occurring at test time.

In the second approach (Section 3.2) we also train on different tasks, but represent the tasks *explicitly* in the input to the model. In this way, the model can learn which predictions lead to low decision errors on which tasks. The goal of this approach is to train a model that can focus its predictions to specific tasks, improving the overall task generalization performance.

## 3.1 IMPLICIT TASK GENERALIZATION

One way to improve the model's ability to generalize over changing tasks, is simply to train it – using an existing DFL method – on many tasks. In terms of feature-target pairs $(x, c)$ to learn over, we are limited to the data available in dataset $D$. However, this is not the case for tasks. For example, when generalizing across changing capacities in a knapsack problem, the ground-truth solution for an altered capacity can be computed without requiring additional supervision in the training set. Hence, in a gradient descent procedure, we can sample a new task $t$ in the forward pass, compute ground-truth solution $z_t^\star(c)$ and 'predicted' solution $z_t^\star(m_\omega(x))$, and backpropagate the task loss.

This approach is outlined in Algorithm 1. In line 4, a new task $t$ is sampled from some distribution. In this work, we do not consider out-of-distribution generalization, and thus sample $t$ from the true task distribution $\mathcal{T}$. In line 5, the ground-truth optimal solution with respect to task $t$ is computed. In line 6, a cost vector prediction $\hat{c}$ is made by the predictive model. For the implicit approach, the model makes its predictions only on the basis of contextual features $x$, i.e., $\hat{c} = m_\omega(x)$. In line 7, the optimal solution of task $t$ for the predicted cost vector is computed. In line 8, the task loss – in this case, the regret – is computed. Finally, in line 9, the task loss is backpropagated in order to update the parameters $\omega$ of the predictive model, for which an existing DFL method can be used.

---

**Algorithm 1** Gradient descent in decision-focused learning with task generalization

---

**Input**: Training data D $\equiv \{(x_i, c_i)\}_{i=1}^N$;
**Hyperparams**: $\alpha$: learning rate, $epochs$: number of epochs

1: Initialize $\omega$
2: **for** each epoch **do**
3:     **for** each instance $(x, c)$ **do**
4:         Sample a task $t = (A_t, b_t)$ from $\mathcal{T}$
5:         $z_t^\star(c) = \arg\min_{A_t z \leq b_t} c^\top z$                     ▷ Compute ground-truth solution
6:         $\hat{c} = m_\omega(x, t)$                             ▷ Compute predicted parameters
7:         $z_t^\star(\hat{c}) = \arg\min_{A_t z \leq b_t} \hat{c}^\top z$             ▷ Compute predicted solution
8:         $\mathcal{L}_t = c^\top z_t^\star(\hat{c}) - c^\top z_t^\star(c)$              ▷ Compute task loss
9:         $\omega \leftarrow \omega - \alpha \frac{d\mathcal{L}_t}{dz_t^\star(\hat{c})} \frac{dz_t^\star(\hat{c})}{d\hat{c}} \frac{d\hat{c}}{d\omega}$      ▷ Backpropagate task loss
10:     **end for**
11: **end for**

---

## 3.2 EXPLICIT TASK GENERALIZATION

With the explicit generalization approach, the intention is that by representing the task explicitly in the input, the model will be able to focus its predictions to specific tasks. With this approach, we shift from a model $m_\omega(x)$, to a model $m_\omega(x, t)$, where $t = (A_t, b_t)$.

To represent an (I)LP, we start by converting the problem to a bipartite variable-constraint graph with additional variable, constraint and edge features, as introduced in (Gasse et al., 2019). In that paper, they use this representation to learn branch-and-bound variable selection heuristics for solving ILPs. Because in the present work the coefficients in the optimization problem's objective function are unknown, our graph representation of the problem only represents its constraints. Consider an (I)LP $P$ with constraints $Az \leq b$, where $A \in \mathbb{R}^{m \times n}$ and $b \in \mathbb{R}^m$ (see equation 1c). This (I)LP can be represented by a bipartite graph $\mathcal{B} = (\mathcal{V}, \mathcal{C}, \mathcal{E})$ and functions $f$, $g$ and $h$. Node set $\mathcal{V} = \{v_i\}_{i=1}^n$ contains a node for each variable in $P$. Similarly, node set $\mathcal{C} = \{C_j\}_{j=1}^m$ contains a node for each constraint in $P$. Set $\mathcal{E} = \{(v_i, C_j) \mid A_{ij} \neq 0\}$ contains the edges connecting variables to the constraints they appear in (i.e., in which they have a non-zero coefficient). Finally, functions $f : \mathcal{V} \to \mathbb{R}^d$, $g : \mathcal{C} \to \mathbb{R}^p$ and $h : \mathcal{E} \to \mathbb{R}^l$ respectively map variable nodes, constraint nodes and edges onto feature vectors.

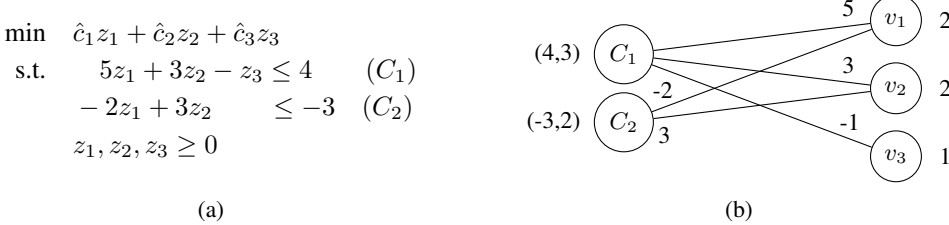

Figure 1: (a) An example LP (b) representation of the constraints by a bipartite graph.

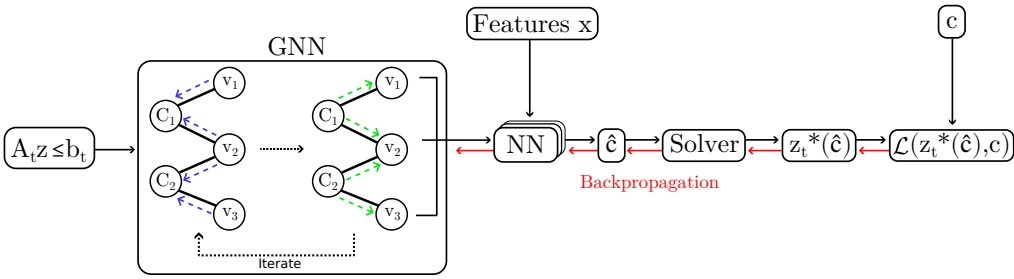

Figure 2: A forward and backward pass using explicit generalization.

In the present work, $f$, $g$ and $h$ are kept simple. We use the same features as (Khalil et al., 2022) (which uses this representation to aid in the *solving* of ILPs, like (Gasse et al., 2019)); excluding the objective coefficients, as they are unknown in our setting. Concretely, function $f$ maps a variable node $v_i$ onto its degree, i.e., the number of constraint nodes it is connected with. Function $g$ maps a constraint node $C_j$ onto its degree and its corresponding constraint right-hand side value $b_j$. Lastly, function $h$ maps edge $e_{ij} = (v_i, C_j)$ onto constraint coefficient $A_{ij}$. Figure 1 shows an example LP (1a) and its bipartite graph representation (1b).

Our explicit task generalization approach is illustrated in Figure 2. The figure represents one forward and backward pass through model $m_\omega(x, t)$ (lines 6 to 9 in Algorithm 1), where $x$ are the contextual features, and $t = (A_t, b_t)$ is the task sampled in this training iteration (line 4). The method starts by constructing the variable-constraint graph $\mathcal{B}$. This graph is then given to a message-passing GNN, which starts by encoding the variable features $f(v_i)$, constraint features $g(C_i)$ and edge features $h(v_i, C_j)$ as embeddings in $\mathbb{R}^k$. We respectively denote these embeddings as $\mathbf{v}_i^0$, $\mathbf{C}_j^0$ and $\mathbf{e}_{ij}^0$. The GNN then performs $L$ iterations of message-passing from variable nodes to constraint nodes and vice versa, where each $l$-th variables-to-constraints pass and each $l$-th constraints-to-variables pass respectively performs the following updates:

$$\mathbf{C}_j^{l+1} = f_{merge}^l(\mathbf{C}_j^l, f_{aggr}(\{\phi(\mathbf{v}_i^l, \mathbf{e}_{ij}^l) \mid v_i \in \mathcal{N}(C_j)\})) \tag{6}$$

$$\mathbf{v}_i^{l+1} = f_{merge}^l(\mathbf{v}_i^l, f_{aggr}(\{\phi(\mathbf{C}_j^{l+1}, \mathbf{e}_{ij}^l) \mid C_j \in \mathcal{N}(v_i)\})) \tag{7}$$

Each layer $l$ has an edge encoder that constructs $\mathbf{e}_{ij}^l$ from $h(v_i, C_j)$. Function $\phi$ is used to construct a message in $\mathbb{R}^k$ from $\mathbf{v}_i$ and $\mathbf{e}_{ij}$. Aggregation function $f_{aggr}$ (e.g., sum, mean or max) aggregates incoming messages in a node. Finally, $f_{merge}^l$ is a function (e.g., a linear model) that takes a node's current embedding, and merges it with the aggregated incoming messages to that node.

After $L$ message passing iterations, the variable nodes $v_i$ have gotten their final variable node embeddings $\mathbf{v}_i^L$. Each node embedding $\mathbf{v}_i^L$ is then separately passed, alongside contextual features $x$, to a predictive model, which predicts objective coefficient $\hat{c}_i$. This could be a single neural network that is shared for each $\hat{c}_i$, or a collection of neural networks, each of which is dedicated to a single $\hat{c}_i$. The entire resulting cost vector $\hat{c}$ is then passed to the solver, which computes solution $z_t^\star(\hat{c})$. Finally, loss $\mathcal{L}_t(z_t^\star(\hat{c}, c))$ is computed, and backpropagated to update both the MLP and the GNN.

## 4 EXPERIMENTS

In this section, we will experimentally answer the following research questions:

**Q1**: How well does state-of-the-art DFL method SPO generalize across changing tasks?

**Q2**: Does training on many tasks sampled at training time (i.e., implicit generalization) improve generalization performance?

**Q3**: Does the explicit representation of the optimization tasks in the input to the predictive model (i.e., explicit generalization) offer an additional improvement?

### 4.1 EXPERIMENTAL SETUP

The prediction-focused method aims to minimize the mean squared error (MSE) of $\hat{c}$ with respect to $c$. As decision-focused method, we opt for the SPO method (Elmachtoub & Grigas, 2022), as it a seminal DFL approach, and has shown great performance across the board in comparative analyses (Tang & Khalil, 2022a; Mandi et al., 2023). The model trained using regular SPO (i.e., without generalization at training time) is always trained on a *base task*. The models trained on different tasks (denoted by 'SPOImplGen' and 'SPOExplGen') are instead trained on tasks sampled from a distribution $\mathcal{T}$ that will be specified for each problem in the results section. Test set performance will be reported for both the base task, and for 50 additional tasks sampled from $\mathcal{T}$ at test time. For each seed, all methods' generalization performance is computed with respect to the same 50 random tasks. The differences between the tasks in $\mathcal{T}$ can take various forms. In this experimental evaluation, we will consider changes in the $A$ matrix or $b$ vector (equation 1), and the exclusion of decision variables from the optimization problem. All code will be made available upon acceptance of the paper.

**Optimization problems.** We experiment on three optimization problems: the 0-1 knapsack problem with unknown item values, the capacitated facility location problem with unknown facility costs and transportation costs, and the undirected shortest path with unknown edge costs. The exact formulation of these optimization problems can be found in appendix A.

**Ground-truth mapping.** For the ground-truth mapping between features and costs, we use the synthetic benchmark from (Elmachtoub & Grigas, 2022), which is commonly used in evaluations of DFL methods (Tang & Khalil, 2022a; Mandi et al., 2022; 2023). In this benchmark, the mapping is a polynomial kernel of degree $deg$. We use $deg = 6$. Details can be found in appendix B. For each dataset, 1000 $(x, c)$ pairs are generated, and split into train, validation and test sets with respective proportions 0.8, 0.1 and 0.1. In each experiment, we run each method on three different ground-truth mappings with varying seeds, and report the averaged results and standard errors.

**Predictive model.** The $deg$ parameter was originally intended to specify the degree of *model misspecification* (Elmachtoub & Grigas, 2022). That is, if a linear predictive model is used, a higher value of $deg$ means that the model is more misspecified. This was done to be able to assess the ability of DFL methods to lead to good decisions when the predictive model is not sufficiently complex to capture the true relationship between features and cost coefficients. Since the benchmark was devised with this in mind, we stick to the use of linear predictive models (i.e., the NNs in Figure 2 are linear models in our experiments). This means that, regardless of whether explicit task representations are used, the features $x$ only go through linear activations. A separate linear model is used for each $\hat{c}_i$ in $\hat{c}$.

**GNN architecture.** In the GNN used to build task representations, the variable, constraint and edge encoders are all 2-layer MLPs with ReLU activation. We used embeddings of size 16. Two message passing layers were used, meaning that the GNN performs two variable-to-constraint passes (equation 6) and two constraint-to-variable passes (equation 7). Function $\phi$ builds a message from a variable or constraint node embedding and an edge embedding by summing up the embedding vectors and performing a ReLU activation on the result. Aggregation function $f_{aggr}$ takes the mean of the incoming messages. Finally, merge function $f_{merge}$ takes in the node's current embedding and the aggregation of the incoming messages, and produces the node's new embedding through a linear model. Afterwards, the merged embedding is normalized to unit length. Between the two message-passing iterations, all node embeddings go through a ReLU activation. The GNN's trainable components are its encoders and the linear model used for merging.

Table 1: Relative regrets on both the base knapsack problem (base regret) and on knapsack problems with altered constraints (generalization regret) for various methods.

| Method | Base regret | Gen. regret | Base regret | Gen. regret | Base regret | Gen. regret |
|---|---|---|---|---|---|---|
| MSE | $2.83\% \pm 0.431$ | $2.91\% \pm 0.064$ | $2.83\% \pm 0.449$ | $3.04\% \pm 0.07$ | $2.91\% \pm 0.426$ | $3.85\% \pm 0.116$ |
| SPO | $0.03\% \pm 0.002$ | $0.04\% \pm 0.001$ | $\mathbf{0.03\% \pm 0.003}$ | $0.08\% \pm 0.005$ | $\mathbf{0.03\% \pm 0.003}$ | $0.45\% \pm 0.046$ |
| SPOImplGen | $\mathbf{0.02\% \pm 0.002}$ | $\mathbf{0.03\% \pm 0.001}$ | $\mathbf{0.03\% \pm 0.004}$ | $\mathbf{0.04\% \pm 0.002}$ | $0.08\% \pm 0.022$ | $0.20\% \pm 0.02$ |
| SPOExplGen | $0.04\% \pm 0.004$ | $0.04\% \pm 0.001$ | $0.04\% \pm 0.007$ | $\mathbf{0.04\% \pm 0.002}$ | $0.07\% \pm 0.009$ | $\mathbf{0.17\% \pm 0.016}$ |
| | Rel. capacity $\sim U(0.4, 0.6)$ | | Rel. capacity $\sim U(0.25, 0.75)$ | | Rel. capacity $\sim U(0, 1)$ | |
| | | | Knapsack capacity generalization | | | |
| MSE | $2.57\% \pm 0.351$ | $2.71\% \pm 0.054$ | $2.56\% \pm 0.428$ | $2.82\% \pm 0.047$ | $2.92\% \pm 0.502$ | $2.94\% \pm 0.054$ |
| SPO | $\mathbf{0.04\% \pm 0.005}$ | $0.35\% \pm 0.009$ | $\mathbf{0.05\% \pm 0.004}$ | $1.98\% \pm 0.048$ | $\mathbf{0.03\% \pm 0.004}$ | $3.27\% \pm 0.143$ |
| SPOImplGen | $0.10\% \pm 0.004$ | $0.09\% \pm 0.002$ | $0.40\% \pm 0.084$ | $0.35\% \pm 0.011$ | $0.53\% \pm 0.070$ | $0.50\% \pm 0.018$ |
| SPOExplGen | $0.10\% \pm 0.011$ | $\mathbf{0.08\% \pm 0.002}$ | $0.12\% \pm 0.012$ | $\mathbf{0.14\% \pm 0.003}$ | $0.15\% \pm 0.008$ | $\mathbf{0.18\% \pm 0.006}$ |
| | $w \sim N(1, 0.25)$ | | $w \sim N(1, 0.5)$ | | $w \sim N(1, 0.75)$ | |
| | | | Knapsack weight vector generalization | | | |
| MSE | $2.87\% \pm 0.461$ | $2.90\% \pm 0.072$ | $2.93\% \pm 0.479$ | $3.13\% \pm 0.078$ | $2.89\% \pm 0.451$ | $4.14\% \pm 0.158$ |
| SPO | $\mathbf{0.03\% \pm 0.002}$ | $0.10\% \pm 0.008$ | $\mathbf{0.04\% \pm 0.006}$ | $0.36\% \pm 0.03$ | $\mathbf{0.03\% \pm 0.003}$ | $0.69\% \pm 0.041$ |
| SPOImplGen | $\mathbf{0.03\% \pm 0.006}$ | $0.10\% \pm 0.008$ | $0.06\% \pm 0.01$ | $0.14\% \pm 0.011$ | $0.14\% \pm 0.014$ | $\mathbf{0.34\% \pm 0.016}$ |
| SPOExplGen | $\mathbf{0.03\% \pm 0.002}$ | $\mathbf{0.06\% \pm 0.004}$ | $\mathbf{0.04\% \pm 0.008}$ | $\mathbf{0.11\% \pm 0.010}$ | $0.21\% \pm 0.058$ | $0.43\% \pm 0.019$ |
| | Subset size 32 | | Subset size 20 | | Subset size 8 | |
| | | | Knapsack item subset generalization | | | |

**Training** To train the linear predictive model (and the GNN in the case of explicit task representations), the ADAM optimizer is used (Kingma & Ba, 2014). Through tuning on separate datasets, a learning rate of $0.001$ has been determined for all methods. When explicit task representations are used, a learning rate of $0.0005$ is used to train the GNN. All models are trained for 100 epochs, as this was sufficient for all models to reach convergence during tuning.

## 4.2 RESULTS

We first describe the structure of Table 1 (knapsack), Table 2 (facility location) and Table 3 (shortest path). We then interpret the results with respect to the research questions in the next section.

Table 1 shows the results for the 0-1 knapsack problem. It contains 40 items (i.e., $d = 40$). In the base task, each ground-truth item weight was sampled from $N(1, 0.75)$. The ground-truth capacity is half of the sum of the item weights. We vary the type of generalization and the similarity between tasks: in the top block, the relative *capacity* is varied (a change in vector $b$), and task distribution $\mathcal{T}$ contains knapsack problems where the capacity is $R$ times the sum of the item weights, where $R$ is uniformly sampled between $l$ and $u$. The 3 column groups have an increasingly wide range of $l$ and $u$, i.e., more variance. In the middle block, the *weight vector* is changed (a change in matrix $A$), and sampled from $N(1, \sigma)$, with an increase in sigma over the column groups. In the bottom block, the knapsack problems do not contain all items, but a random *subset* of the items (the exclusion of decision variables). Each resulting problem has a capacity that is half of the sum of the weights of the items in the associated subset. The subset size becomes smaller over the column groups.

Table 2 contains similarly presented results for the facility location problem. It contains 10 customers and 5 facilities (i.e., $d = 55$: 5 fixed facility selection costs and 50 transportation costs). The base problem's ground-truth capacities are sampled from $N(10, 5)$. Each ground-truth customer demand is sampled from $N(2.5, 1.25)$. In the top table, the *capacities* are altered (a change in matrix $A$), and (re)sampled from $N(10, \sigma)$, for increasingly large $\sigma$ per group. In the bottom table, the *demands* are changed (also a change in matrix $A$), and (re)sampled from $N(2.5, \sigma)$, for varying $\sigma$.

Table 3 shows the results for the shortest path problem over a $5 \times 5$ lattice (as in (Elmachtoub & Grigas, 2022)). The base problem uses fixed start and end nodes in the bottom-left and top-right of the lattice. The altered tasks contain different start and end nodes (a change in vector $b$), sampled uniformly at random from the set of nodes.

Table 2: Relative regrets on the both the base facility location problem (base regret) and on facility location problems with altered constraints (generalization regret) for various methods.

| Method | Base regret | Gen. regret | Base regret | Gen. regret | Base regret | Gen. regret |
|---|---|---|---|---|---|---|
| MSE | $29.71\% \pm 17.19$ | $27.72\% \pm 1.945$ | $31.18\% \pm 18.485$ | $30.12\% \pm 2.166$ | $30.61\% \pm 19.059$ | $25.95\% \pm 2.014$ |
| SPO | $1.8\% \pm 0.703$ | $6.53\% \pm 0.376$ | $1.54\% \pm 0.401$ | $57.33\% \pm 6.158$ | $2.04\% \pm 0.774$ | $80.07\% \pm 9.382$ |
| SPOImplGen | $5.07\% \pm 0.752$ | $2.49\% \pm 0.174$ | $3.84\% \pm 0.988$ | $3.48\% \pm 0.141$ | $3.14\% \pm 1.341$ | $3.23\% \pm 0.159$ |
| SPOExplGen | $\mathbf{1.21\% \pm 0.169}$ | $\mathbf{1.03\% \pm 0.041}$ | $\mathbf{1.12\% \pm 0.301}$ | $\mathbf{1.35\% \pm 0.052}$ | $\mathbf{1.29\% \pm 0.289}$ | $\mathbf{1.56\% \pm 0.073}$ |
| | Capacities $\sim N(10, 2.5)$ | | Capacities $\sim N(10, 5)$ | | Capacities $\sim N(10, 7.5)$ | |

Facility location capacity generalization

| Method | Base regret | Gen. regret | Base regret | Gen. regret | Base regret | Gen. regret |
|---|---|---|---|---|---|---|
| MSE | $31.31\% \pm 18.611$ | $30.57\% \pm 2.604$ | $31.14\% \pm 18.43$ | $32.48\% \pm 2.769$ | $30.19\% \pm 17.544$ | $29.68\% \pm 2.625$ |
| SPO | $1.39\% \pm 0.412$ | $2.2\% \pm 0.123$ | $1.53\% \pm 0.393$ | $13.77\% \pm 1.544$ | $1.84\% \pm 0.543$ | $15.87\% \pm 1.539$ |
| SPOImplGen | $1.69\% \pm 0.505$ | $1.84\% \pm 0.093$ | $1.55\% \pm 0.335$ | $2.45\% \pm 0.23$ | $1.91\% \pm 0.394$ | $2.48\% \pm 0.17$ |
| SPOExplGen | $\mathbf{0.68\% \pm 0.241}$ | $\mathbf{0.9\% \pm 0.044}$ | $\mathbf{0.72\% \pm 0.209}$ | $\mathbf{1.45\% \pm 0.24}$ | $\mathbf{0.88\% \pm 0.279}$ | $\mathbf{1.63\% \pm 0.17}$ |
| | Demands $\sim N(2.5, 0.625)$ | | Demands $\sim N(2.5, 1.25)$ | | Demands $\sim N(2.5, 1.875)$ | |

Facility location demand generalization

Table 3: Relative regrets on the both the base shortest path problem (base regret) and on shortest path problems with altered start and end nodes (generalization regret) for various methods.

| Method | Base regret | Gen. regret |
|---|---|---|
| MSE | $11.94\% \pm 1.652$ | $15.53\% \pm 1.057$ |
| SPO | $4.49\% \pm 0.174$ | $42.31\% \pm 4.006$ |
| SPOImplGen | $4.73\% \pm 0.286$ | $\mathbf{10.15\% \pm 0.701}$ |
| SPOExplGen | $4.42\% \pm 0.623$ | $\mathbf{10.10\% \pm 0.674}$ |

Shortest path generalization

## 4.3 DISCUSSION

**Q1: Generalization performance of SPO.** Across the board, SPO outperforms MSE on the base task it has been trained on. However, when the task is changed at test time, the SPO model's performance degrades. This effect increases when the variance of task distribution $\mathcal{T}$ increases (column groups more towards the right). For variations in the weights of knapsack problems or in the capacities of facility location problems, or for changing start and end nodes in the shortest path, the predictions of the model trained with SPO can even become counter-productive, meaning that they lead to *worse decisions* than the ones made by the prediction-focused MSE model. SPO clearly does not generalize well to new tasks. This makes sense: the predictive trade-offs that allow the SPO model to perform well on the specific task it has been trained on may be the wrong trade-offs for altered tasks. For example, in the knapsack problem, the model trained by SPO may learn to systematically overestimate the value of items that should be selected (due to low item weights, for example) to ensure the selection of these items. But those same overestimations may lead to the wrong decisions being taken when those items are assigned a large weight at test time instead.

**Q2: Implicit task generalization.** Implicit generalization (i.e., training across many tasks, sampled at training time) significantly improves task generalization performance across the board. While performance still worsens as the task distribution $\mathcal{T}$ widens, it worsens significantly less than SPO did. It turns out that training on many tasks still outperforms the prediction-focused MSE approach by a large margin, even for the widest distributions $\mathcal{T}$ considered (right-most column groups). This means that while the model is trained on many tasks from $\mathcal{T}$, these tasks still have enough in common for the model to learn to predict in a way that improves overall decision quality over $\mathcal{T}$ compared to prediction-focused learning. For instance, for weight generalization in the knapsack problem, the model might still learn to overestimate items with large cost coefficients, since across the different weight vectors in $\mathcal{T}$, these items should on average still be selected more often than items with small cost coefficients. However, implicit generalization generally does not perform as well as SPO does

*on the base task*, meaning that by training on many tasks implicitly, something is sacrificed: the resulting model does well on average across $\mathcal{T}$, but does not excel on specific tasks.

**Q3: Explicit task generalization**  The effect of explicit generalization seems to vary between the optimization problems and types of task generalization. For weight generalization in the knapsack problem, the explicit task representations provide a significant additional benefit over the implicit generalization approach, and this performance gap grows as task distribution $\mathcal{T}$ widens. Also for demand and capacity generalization in the facility location problem, a large improvement is realized by the use of task representations, *even for narrow task distributions*. These three experiments have in common that the task alterations take place in the left-hand side of the constraints, i.e., in the $A$ matrix of the $Az \leq b$ constraint. This matrix contains the coefficients of the decision variables in the constraints, which are represented as edge features in the bipartite graph representation given to the GNN. When the changes take place in the $b$ vector (as in shortest path start and end node generalization and knapsack capacity generalization), or involve the exclusion of decision variables (as in knapsack subset generalization), the explicit approach offers less, or even no benefit.

Interestingly, for both demand and capacity generalization on the facility location problem, explicit generalization even significantly outperforms regular SPO *on the base task*, meaning that it did not merely avoid the degradation of performance when the task changes, but even *improved* the ability to predict for a specific task $t \in \mathcal{T}$. Existing DFL work does not explicitly represent the optimization problem in the input to the predictive model. The only way the model can learn to adapt its predictions to the task at hand, is through the incoming gradients of the task loss. We observe that, through explicit representation of the task, the dependencies between the decision variables, and hence the predictions, can be better captured. The facility location problem, for example, contains more complex structure than the knapsack and shortest path problems. There are two types of decision variables whose coefficients are predicted: facility selection variables, and transportation variables. These variables are strongly connected: if a facility is not selected, the transport from this facility to each customer is necessarily zero. We hypothesize that it is this comparatively complex structure that leads the use of task representations to do so well on this benchmark.

## 5  CONCLUSION AND FUTURE WORK

This paper investigated the ability of DFL models to generalize across changing optimization tasks, and proposed two approaches to improve in this regard. We introduced a new learning objective, in which the goal is not to learn to make predictions that lead to good decisions on a single optimization task, but across a *distribution* of tasks. In our experiments, we showed that the state-of-the-art SPO approach leads to predictions that perform very well on the task it has been trained on, but that quickly lead to deteriorating performance when the task changes.

To improve on this, we first proposed implicit generalization, which trains the model on many tasks sampled from the task distribution. Experiments showed that implicit generalization greatly improved the average performance of the resulting model across the task distribution, but did not lead to models that excel at any specific task, in the way that a model trained by SPO does.

To reduce this gap, we proposed explicit task generalization, which also trains on many tasks, but involves explicit task representations in the input to the predictive model. The task representations are built by a GNN that operates on the bipartite variable-constraint graph representing the optimization task's constraints. Our experiments showed that these task representations create a significant additional improvement in performance over implicit generalization when the task alterations occur in the left-hand side of the constraints. Remarkably, on the most complex benchmark, the explicit generalization model even surpassed the performance of the SPO model on the very task on which the latter was specifically trained. This finding underscores the potential of task representations to elevate model performance, even in comparison to models tailored to individual tasks.

In future work, we want to expand on the final finding discussed above, and further investigate the advantages task representations can bring on individual optimization tasks. We will also investigate another type of task generalization, in which the task grows, and additional variables are added. Such generalization has promise not only because it would be able to deal with another realistic type of task alteration, but also because it could lead to a new way of improving the *scalability* of DFL, by training models on small tasks, but later, at test time, using them for larger ones.

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

## A OPTIMIZATION PROBLEMS

**Knapsack Problem.** Given that there is a set of items $\mathcal{I} = \{1, \ldots, n\}$ where item $i \in \mathcal{I}$ has a weight $w_i$ and value $c_i$, the knapsack problem is to find the subset of all items with maximum total value without exceeding the knapsack capacity $W$. Let

$$z_i = \begin{cases} 1 & \text{if item } i \text{ is selected} \\ 0 & \text{otherwise} \end{cases}$$

for all $i \in \mathcal{I}$. The mathematical model for the knapsack problem is given by

$$\max \quad \sum_{i \in \mathcal{I}} c_i z_i \tag{8}$$

$$\text{s.t.} \quad \sum_{i \in \mathcal{I}} w_i z_i \leq W \tag{9}$$

$$z_i \in \{0, 1\} \text{ for all } i \in \mathcal{I}. \tag{10}$$

The objective function (8) maximizes the total value of selected items. Constraint (9) ensures that the knapsack capacity is not exceeded. Constraint (10) is the domain constraint for the decision variables.

**Capacitated Facility Location Problem.** Given a set of potential facility locations $\mathcal{I} = \{1, \ldots, n\}$ and a set of customers $\mathcal{J} = \{1, \ldots, m\}$, the capacitated facility location problem is to minimize the total operation cost while satisfying the customer demands of a single product. Let $d_j$ be the demand of customer $j$, $f_i$ be the fixed cost of opening a facility at location $i$, $c_{ij}$ be the transportation cost for one unit of product from a facility located at $i$ to customer $j$, $u_i$ be the capacity of a facility located at $i$. Also, let

$$z_i = \begin{cases} 1, & \text{if a facility is located at } i \\ 0, & \text{otherwise} \end{cases}$$

and $y_{ij} =$ the fraction of demand of customer $j \in \mathcal{J}$ from a facility located at $i \in \mathcal{I}$ be the decision variables. Then, the following optimization model solves the capacitated facility location Problem

$$\min \quad \sum_{i \in \mathcal{I}} \sum_{j \in \mathcal{J}} c_{ij} d_j y_{ij} + \sum_{i \in \mathcal{I}} f_i z_i \tag{11}$$

$$\text{s.t.} \quad \sum_{i \in \mathcal{I}} y_{ij} = 1 \text{ for all } j \in \mathcal{J} \tag{12}$$

$$\sum_{j \in \mathcal{J}} d_j y_{ij} \leq u_i z_i \text{ for all } i \in \mathcal{I} \tag{13}$$

$$z_i \in \{0, 1\}, y_{ij} \geq 0 \text{ for all } i \in \mathcal{I} \text{ and } j \in \mathcal{J}. \tag{14}$$

The objective (11) minimizes the total cost of transportation and fixed costs for the opened facilities. Constraint (12) ensures that the demand of each customer is satisfied. Constraint (13) guarantees that facility capacities are respected and only the opened facilities are used. Note that, while the capacity $u_i$ is written on the right-hand side of the constraint, it is actually part of left-hand side matrix $A$ because it is associated with a decision variable, which can be seen when the problem is written in the standard form (as in equation 1). Constraint (14) is the domain constraint for the decision variables.

**Shortest Path Problem.** Let $\mathcal{G} = (\mathcal{N}, \mathcal{A})$ be a graph with the set of nodes $\mathcal{N}$ and the set of arcs $\mathcal{A}$. Also, let $s$ and $t$ in $\mathcal{N}$ be the two specialized nodes in $\mathcal{N}$ corresponding to source and destination, respectively. For each arc $(i, j) \in \mathcal{A}$, a nonnegative cost $c_{ij}$ is given. The shortest path problem is to find a route from $s$ to $t$ with the least total arc costs. If we define

$$z_{ij} = \begin{cases} 1 & \text{if arc } (i, j) \text{ is on the path} \\ 0 & \text{otherwise} \end{cases}$$

for all $(i, j) \in \mathcal{A}$, the following mathematical model solves the shortest path problem.

$$\min \quad \sum_{(i,j) \in \mathcal{A}} c_{ij} z_{ij} \tag{15}$$

$$\text{s.t.} \quad \sum_{j:(i,j) \in \mathcal{A}} z_{ij} - \sum_{j:(j,i) \in \mathcal{A}} z_{ij} = \begin{cases} 1, & \text{if } i = s \\ -1, & \text{if } i = t \\ 0, & \text{otherwise} \end{cases} \tag{16}$$

$$z_{ij} \in \{0, 1\} \text{ for all } (i, j) \in \mathcal{A}. \tag{17}$$

Objective (15) minimizes the total cost of selected arcs. Constraint (16) ensures that the route starts from the source node $s$ and ends at the destination node $t$. Constraint (17) is the domain constraint for the decision variables. Alternatively, the Bellman-Ford algorithm can be used to solve the shortest path problem even when the costs of some edges are negative.

## B    GENERATION OF THE GROUND-TRUTH MAPPING

This section specifies the generation of the ground-truth mapping between features $x$ and cost coefficients $c$ used in the experiments. This data generation process is taken from (Elmachtoub & Grigas, 2022), and is often used in evaluations of DFL works Tang & Khalil (2022a); Mandi et al. (2022; 2023).

First, a matrix $Q \in \mathbb{R}^{5 \times p}$, where $p$ is the dimension of the cost vector to be predicted (which varies between the benchmarks). $Q$ contains the parameters of the ground-truth model. Each entry of $Q$ is sampled from a Bernoulli distribution that takes value 1 with probability 0.5, and value 0 otherwise.

Then, for each instance $i$ to be generated, feature vector $x_i \in \mathbb{R}^5$ is sampled from the standard normal distribution $N(0, I_5)$. From $x_i$ and $Q$, the cost vector $c_i$ is then generated using a polynomial kernel function as $c_{ij} = (\frac{1}{\sqrt{p}}(Qx_i)_j + 3)^{deg} + 1$. In our experiments, we use $deg = 6$ as the degree of this polynomial relationship.

For the capacitated facility location problem, the fixed costs of opening facilities are generally larger than the transportation costs. Hence, for this problem, we multiply the entries of each $c_i$ associated with a facility cost by 100.

