# OpenReview forum: "Task Generalization in Decision-Focused Learning"
_ICLR.cc/2024/Conference — Submitted to ICLR 2024_

### Official Review · Reviewer_g5nG · 2023-10-24

**Soundness:** 3 good
**Presentation:** 3 good
**Contribution:** 2 fair
**Rating:** 5
**Confidence:** 4

**Summary:**

The paper proposed two methods aimed at improving the generalization performance of the state-of-the-art Decision-Focused Learning (DFL) method SPO. The first proposed method, SPOImplGen, trains the model on different tasks sampled from a task space rather than a task. The second proposed method, SPOExplGen, combines the GNN model with SPOImplGen to acquire a robust task representation, resulting in improved generalization performance. In the experimental section, it is demonstrated that both methods outperform SPO across three different problems in terms of generalization performance.

**Strengths:**

- S1 The paper is well-written and addresses an important topic.
- S2 The proposed methods improve the generalization performance of the SPO, especially on the shortest path problem.
- S3 The paper conducts a thorough analysis of the experiment's results.

**Weaknesses:**

- W1 The settings in this paper need further justification, although they have been adopted in other existing works.
  - W1-1 Given the primal-dual relationship for LP, $c$ and $A$ serve a similar role but only $c$ is assumed to be unknown.
  - W1-2 Since c is assumed to be unknown, in which sense we may assume that they are known in the training data? Is there any real-world application that can support such a setting?

- W2 The proposed method is reasonable but not very significant. On the one hand, it is not clear in theory why samples from multiple tasks can improve the generalization performance. On the other hand, while being more complex, SPOExplGen does not demonstrate a significant improvement in generalization performance when compared to SPOImplGen.

- W3 The experiments could be improved in the following ways.
  - W3-1 Since the paper considers samples from multiple tasks, it would be interesting to explore the generalization performance between different instance sizes.
  - W3-2 The problem sizes are relatively small: item size 40 in the knapsack problem, and the 10 customers and 5 facilities in the facility location problem.

**Questions:**

What are the theoretical advantages of the proposed method?

---

> ### Author Response · Authors · 2023-11-17
> **Response part 1**
>
> **Justification for focus on uncertainty in $c$**
>
> Note that, In the dual of an (integer) linear program, the coefficients in the $A$ matrix remain part of the constraints. It is the $b$ vector that moves to the objective, and the $c$ vector that moves to the constraints, when the dual is taken.
>
> But regardless of this, the fact that $A$ remains in the constraints in the dual is not what hinders the application of DFL to problems with uncertainty in the constraints. After all, one could just work with a task loss that expresses the degree to which an $Az \leq b$ constraint is violated, and train the predictive model to minimize that loss instead.
>
> The real reason that this is generally not done is a matter of semantics: a solution that violates the constraints, even by only a little, is infeasible and cannot be used. It must be fixed by means of a recourse action, which comes with an associated cost/penalty. This penalty is often nonlinear or even inexpressible in closed-form (it may be the result of solving yet another optimization problem). This seriously complicates the differentiation of a loss based on the solution after the recourse action with respect to the parameters of the predictive model. It is for this reason that, while there is a very limited body of work that focuses on uncertainty in the constraints, the vast majority of work focuses on uncertainty in the objective.
>
>
> **Can $c$ be assumed to be present in training data**
>
> In real-world predict-then-optimize applications, $c$ is typically uncertain at solving time, but revealed later on. For instance, consider a production company that must schedule its production processes based on uncertain future energy prices (which take the role of $c$), which are to be predicted based on correlated contextual features. Although the energy prices are uncertain at decision-making time, they will become known in due time, and can thus become part of a set of historical training data. Similarly, consider a delivery company that must make its routing decisions based on future traffic conditions (which again take the role of $c$). Although the traffic conditions are not yet known when the routes are to be computed, in time, they will become known.
>
>
> **Why the methods work**
>
> While the merits of our work are mostly evidenced empirically, and while a theoretical analysis was outside of the scope of this paper, there is certainly a strong intuitive explanation of why training on multiple tasks improves generalization performance. When a predictive model is trained on a single task, the only aim in training is to achieve the best possible performance on that task. While this is commonly done in existing DFL literature, we show that such models perform badly on altered optimization tasks. When, instead, training implicitly on a distribution of tasks, it is the average performance across this distribution that matters, and thus the trained models tend to have a more robust generalization performance. However, because the predictive trade-offs required to perform well on one task may run counter to the trade-offs required for another task – and because these models do not know which task they are predicting for at any point in time – they generally are not able to excel on individual tasks. So, to be able to excel on individual tasks, the model must receive some knowledge about the current task, which is where task representations and explicit generalization come in.
>
> A further theoretical analysis of the task generalization problem and the proposed approaches for tackling it would definitely strengthen this line of work. But given the 9-page limit of this conference paper, we considered this out of scope.
>
>
> **The benefit of explicit generalization over implicit generalization**
>
> There are indeed cases where explicit generalization does not offer a large benefit over implicit generalization. However, the claim that SPOExplGen does not demonstrate a significant improvement in generalization performance when compared to SPOImplGen is false. In many cases, the benefit is large: in knapsack weight generalization, the improvement in regret runs up to 64% (Table 1). In knapsack item subset generalization for large subsets, there is an improvement of 40% (Table 1). In facility location capacity generalization, the improvement goes up to 61% (Table 2), and for facility location demand generalization, it goes up to 51% (Table 2). These are all significant improvements, on top of the improvements that SPOImplGen already achieves with respect to regular SPO.

---

> ### Author Response · Authors · 2023-11-17
> **Response part 2**
>
> **Task generalization between different instance sizes**
>
> Indeed, task generalization across different instances sizes is certainly an interesting setting to investigate. In the present work, we always assume the existence of a ‘base task’ on which the MSE and SPO models are trained. So changing the instance size with respect to the base task could mean two things: making it smaller or making it larger. In our knapsack item subset generalization experiment, we explored the effect of generalizing over smaller instances. Setting up an experiment around generalizing to larger instances is more tricky, however. This is the case because our experimental setup (which was inspired by related work such as (Elmachtoub & Grigas, 2022) and (Mandi et al., 2022, 2023)) involves a separate ground-truth predictive mapping between the features and each decision variable’s cost coefficient. In other words, the mapping from the features to one decision variable’s cost coefficient is different from the mapping from the features to another decision variable’s cost coefficient. The implication of this is that, when a new decision variable appears at test time, no informed prediction can be made for this decision variable’s cost coefficient. So, in order to investigate generalizing over larger tasks, another experimental setup is needed, in which the mapping from features to cost coefficients is shared between all decision variables.
>
> Still, this would be interesting to explore in future work, since a setting in which the predictive mapping is shared across decision variables still reflects many realistic scenarios. Additionally, this kind of generalization is promising from a scalability point of view. As we discuss in our future work section, the ability to effectively train on small tasks, in order to later do inference on larger tasks, can make DFL training significantly faster.
>
>
> **Small problem sizes in experiments**
>
> While it is true that the optimization problems used in the experiments are relatively small, this is the case for most existing DFL work as well (Mandi & Guns, 2020; Mulamba et al., 2021; Elmachtoub & Grigas, 2022; Mandi et al. 2022, 2023; Tang & Khalil, 2022a, Pogancic et al., 2022). The reason for this is that training a model in a decision-focused manner is highly expensive, because for each instance in each epoch, a (potentially NP-hard) optimization problem must be solved. In research papers these problems are typically solved to optimality, restricting experimental settings to ones with relatively small optimization problems. In real-world applications with larger problems, approximate solving may be used instead.

---

> > ### Comment · Reviewer_g5nG · 2023-11-21
> > **Thank you**
> >
> > I appreciate the author's response.

---

### Official Review · Reviewer_nLem · 2023-10-30

**Soundness:** 2 fair
**Presentation:** 2 fair
**Contribution:** 2 fair
**Rating:** 3
**Confidence:** 4

**Summary:**

This paper addresses the task generalization problem in decision-focused learning (DFL) methods. It focuses on the case of integer linear programs (ILPs) and formulates different tasks as variations of the coefficient matrix A and vector b in the constraints. The main technical contribution is to propose two methods to address task generalization in DFL: 1) implicit method which works by simply training DFL on different tasks sampled at training time, and 2) explicit method which maps the A and b to embeddings using GNNs, which are then to be concatenated with the problem features x. The empirical results show that the proposed task generalization methods can improve regular DFL methods on unseen tasks.

**Strengths:**

1) Task generalization seems to be an important aspect of DFL methods (the motivation has not been clearly stated, though)

2) The two methods to do task generalization are heuristic but make sense. In particular, representing tasks as bipartite graphs and then using GNNs to extract the information sounds reasonable. One potential disadvantage is that it introduces a lot more parameters to be learned together with the DFL process.

3) The empirical results seem promising (explicit methods don't have a large edge, though).

**Weaknesses:**

1) The motivation is task generalization is not clear -- why do we care about task generalization in DFL settings? If we have a new task, why don't we start training from scratch? Task generalization in prediction tasks makes more sense, e.g., when you have zero-shot or few-shot data. But here in DFL we don't have such issues.

2) The two methods of task generalization are mostly heuristic, and there is no guarantee or in-depth analysis of whether/why they can work. Task representation using GNNs is not new, either.

3) The empirical results are not entirely convincing and is not reproducible with important setting description missing.
- Not sure if I missed anything, but I did not see where you described how many sampled tasks you need to have to train? Beyond reproduction, this is also very important to understand if the comparison is fair, and to understand what computational overhead is required.
- For the base task, the generalized methods are even better than those trained on the specific base task. This is a bit counter-intuitive. Is it because the generalization takes in more training data? If so, this does not appear a fair comparison.
- The advantage of the explicit generalization is very small compared to the implicit method in Tables 1 and 3. It is not clear how the proposed task representation method works.

4) The writing needs a lot of improvement. E.g.,
- The abstract and introduction spend the majority of the space explaining the motivation/challenges of DFL, which has been there in existing DFL papers. However, it rarely touches on the motivation to do task generalization in DFL, which is the FOCUS of this paper.
- The references are a bit messy. Also, on page 3, Niepart et al. 2021 is mentioned twice under two different ways of handling integer variables.
- Typos. E.g.,
Page 2: Given is;
Page 6, in Figure 2 is are;
Page 7, the caption of Table 1: on the both the base

5) The paper does not have a related work section, and many important DFL papers are not discussed, not even the two earlier and classic DFL papers (where the majority of the ideas discussed in the Introduction are from!):
- Donti et al 2017, Task-based end-to-end model learning in stochastic optimization
- Amos et al 2017, Optnet: Differentiable optimization as a layer in neural networks

**Questions:**

1) In the GNN training, is it trained together with the decision loss and the linear prediction layer? If so, it introduces considerable amount of new parameters and may make the training even more challenging.

2) See my other questions in Weaknesses.

---

> ### Author Response · Authors · 2023-11-17
> **Response part 1**
>
> **Motivation for task generalization**
>
> We, like the other reviewers, do not share the concern that the need for task generalization is insufficiently motivated in the paper, but we are open to hear what the reviewer specifically feels is missing. For reference, the central motivation given in the paper is the following:
>
> > In this work, we identify and address an open problem that has thus far received much less exposition: the limited ability of decision-focused models to generalize across new optimization tasks. Existing DFL methods lead to models that perform well on the optimization problem that they have been trained on, but that generalize poorly to changing tasks, i.e., changing constraints. This is a serious limitation of the state of the art, as real-world optimization problems change all the time. The available resources in scheduling problems, the traversable edges in graph problems and the customer demands in vehicle routing problems, are all subject to constant change.
>
> For the sake of clarity, we want to expand on this here with an example. Consider the following scenario. A delivery company must deliver goods to its customers using delivery trucks. Each day, the company needs to make routing decisions for its trucks, based on the traffic conditions of that day, which are unknown at decision-making time. In order to make routing decisions, a decision-focused predictive model is first used to predict the traffic conditions from known correlated features. In this scenario, it is of vital importance that the predictive model can generalize across decision-making tasks, because each day, there may be different customer demands (both in terms of quantity and location), a different set of inaccessible roads, and a different number of available trucks. Similarly, many other optimization problems that must be solved on a regular basis can be expected to change in unpredictable ways. If the model cannot account for this, and is overfitted on the task it has been trained on, this can lead to highly suboptimal decision-making. It is for this reason that task generalization is of such high importance.
>
> Unfortunately, retraining a model each time the task changes is not a feasible solution to the task generalization problem. This is the case because training models using DFL is computationally highly expensive; much more so than training using traditional prediction-focused methods, as mentioned in the paper. The reason for this is that, in every training iteration, a (potentially NP-hard) optimization problem must be solved. This makes retraining frequently infeasible. Rather, a robust model that can adapt to changing tasks is required, which is what we investigated in this work.
>
> We want to highlight the lack of existing investigation into this topic. Task generalization is a dimension that has thus far received no exposition in the literature on DFL, but that is of vital importance to the broad application of DFL to real-world decision-making.
>
> **Lack of theoretical guarantees**
>
> In this work, we exposed the task generalization issue in DFL, and empirically evaluated two ways of dealing with this issue, with great experimental results. A theoretical analysis of the task generalization issue, and to which degree these methods can be expected to reduce it, would certainly be valuable, but is deliberately kept outside of the scope of this conference paper. To add a meaningful theoretical analysis on top of our empirical evaluation would be hard given the 9-page limit, but would be a great addition to a potential journal extension to this work. Do note, however, that the derivation of theoretical guarantees is rarely present in existing DFL work. A deeper theoretical understanding is certainly important, but presently absent from most related work as well.
>
> **Missing descriptions of number of sampled tasks**
>
> As Algorithm 1 shows, a new task is sampled for each training instance in each epoch, and as detailed in section 4.1, the training set contains 800 instances, on which each model is trained for 100 epochs. This means that each model is trained on a total of 800 * 100 = 80.000 tasks.
>
> Note, also, that training on an arbitrary number of tasks is always possible, and does not require additional supervision from the user: the ground-truth solution $z^\star_t(c)$ of a task $t$ can always be computed, as long as ground-truth cost vector $c$ is known. The baseline SPO method and the generalization methods all use the same training sets of 800 $(x, c)$ pairs, and thus get exactly the same information to work with. They are also trained for the same amount of epochs. Thus, the comparison is fair, regardless of the amount of tasks the generalization methods sample at training time. Also note that the only added computational cost of the generalization methods is the computation of the ground-truth solution of a newly sampled task (line 5 of Algorithm 1).

---

> ### Author Response · Authors · 2023-11-17
> **Response part 2**
>
> **Generalizing models performing better on base task**
>
> The generalizing models generally do not perform better on the base task than the models trained specifically on that task. This can be seen in the ‘base regret’ columns of the tables of results, where the SPOImplGen and SPOExplGen models perform worse than the SPO model (aside from the first column in the knapsack capacity generalization experiment, where the SPOImplGen model does slightly better, which can be attributed to randomness). This is true in all experiments, except for the one on the facility location problem, where the model trained with explicit generalization does notably better on the base task than the SPO model trained specifically on this task. This is certainly not due to training on more tasks; after all, the implicit generalization model trains on the same amount of tasks as the explicit generalization model, and performs worse on the base task than the SPO model trained specifically on that task. Rather, it is due to the use of explicit task representations.
>
> This surprising but highly promising result is discussed in the final paragraph of section 4.2. It highlights the advantage of explicit task representation in DFL: through an explicit representation of the optimization problem in the input to the predictive model, the model can more effectively learn the right predictive trade-offs required to achieve a low task loss on the problem at hand, especially for problems with strong dependencies between the decision variables, such as the facility location problem. This is an interesting finding and can serve as a great starting point for future work into the possible uses of explicit task representation in DFL, as we also highlight in our future work section.
>
>
> **The benefit of explicit generalization over implicit generalization**
>
> As argued in the paper, explicit generalization does not always offer a large benefit over implicit generalization. There are indeed cases where the advantage is small or nonexistent, such as in knapsack capacity generalization (Table 1), knapsack subset generalization for small subsets (Table 1), or start and end node generalization in the shortest path (Table 3). There are, however, also many cases where the benefit is large: knapsack weight generalization (up to 64% improvement, Table 1), knapsack item subset generalization for large subsets (up to 40% improvement, Table 1), facility location capacity generalization (up to 61% improvement, Table 2) and facility location demand generalization (up to 51% improvement, Table 2).
>
> **Double citation**
>
> Thank you for pointing out the double citation of Niepert et al. (2021). The second reference was supposed to be a reference to a different paper. We have fixed this in the revision.
>
> **Lack of a related work section and works that are not discussed**
>
> This work indeed does not have an explicit related work section. However, it does discuss the relevant related work in section 2, which is dedicated to giving the needed background on DFL. We disagree with the statement that many important DFL papers are not discussed. The three major paradigms in solving the zero-gradient issue in DFL are identified and accompanied by several citations. For a more comprehensive overview, the reader is then referred to a survey paper on DFL. Finally, by far the most strongly related work, by Tang & Khalil (2022a), is discussed elaborately. We would be interested to hear which important papers the reviewer feels are missing.
>
> The Amos et al. (2017) and Donti et al. (2017) papers are indeed not discussed. Our motivation for this had to do with the scope of what we considered related work: we limited ourselves to the many works that operate on the same kinds of problems as our work: LPs and ILPs. The Amos et al. (2017) and Donti et al. (2017) papers are instead concerned with continuous nonlinear problems, which come with a separate set of challenges, and which do not suffer from the zero-gradient issue central to most DFL research. For this reason, we did not discuss them. However, because of their seminal status in the literature regarding continuous problems, we added citations to them in the revision.
>
>
> **GNN training**
>
> The GNN is indeed trained end-to-end alongside the linear prediction layer using the task loss, as described at the end of page 5, and illustrated in Figure 1. It is also correct that the use of a GNN introduces additional parameters to be trained. However, we did not observe this making training more challenging. The explicit generalization models (using GNNs) were able to converge in around the same amount of epochs as the other models (not using GNNs) and did not have noticeably less stable learning curves.

---

> > ### Comment · Reviewer_nLem · 2023-11-23
> >
> > Thanks for the responses. I still think that the paper needs significant work to either shed more theoretic insights, or to empirically demonstrate the proposed methods in a more comprehensive and convincing manner. Therefore I will keep my rating.

---

### Official Review · Reviewer_jrAx · 2023-10-30

**Soundness:** 4 excellent
**Presentation:** 4 excellent
**Contribution:** 3 good
**Rating:** 8
**Confidence:** 4

**Summary:**

While the majority of prior research has primarily concentrated on minimizing regret in DFL problems, this paper takes a distinctive approach by emphasizing the crucial aspect of generalization capability, particularly in the context of different tasks. To achieve this, the authors conduct a thorough examination of the well-established method for DFL, namely SPO.

The evaluation of generalization is undertaken by introducing variations in task-specific inputs (e.g. altering source or destination nodes within a graph), and subsequently assessing the performance of the DFL method. The findings reveal a noteworthy observation: while SPO excels in terms of normalized regret, its performance experiences a decline when tasked with adapting to different tasks.

In response to this observation, the paper introduces two highly effective mechanisms to address this issue. Firstly, an implicit strategy involves the random sampling of distinct task instances at each update. Secondly, an explicit approach entails encoding I(LP) instance using a graph neural network, seamlessly integrating it into the DFL training pipeline. These adaptations significantly enhances the generalization performance, as demonstrated in the experiments section.

**Strengths:**

The experiments conducted in this study provide compelling evidence. They demonstrate that the state-of-the-art SPO approach exhibits exceptional performance on the specific task for which it was trained. However, a noteworthy observation emerges: this performance rapidly deteriorates when confronted with a change in task. This meticulous analysis of SPO's generalization capabilities represents a commendable and significant contribution to the field. To the best of my knowledge, this facet has not been extensively explored in prior research.

Furthermore, the proposed mechanisms designed to enhance generalization stand out as a pivotal advancement in the field. They fortify the adaptability and resilience of Decision focused learning (DFL) across a spectrum of task settings. The rationale behind the effectiveness of both mechanisms is articulated clearly and substantiated with empirical evidence, as vividly elucidated in Section 4.

**Weaknesses:**

- There has been recent advance in the area of parametric surrogate learning [1,2]. Authors in those papers also capitalize on generalization property of the learned surrogates. I'd appreciate authors' discussion and analysis on this matter, possibly including some comparison.

[1] S. Shah, K. Wang, B. Wilder, A. Perrault, and M. Tambe. Decision-focused learning without decision-making: Learning locally optimized decision losses. In NeurIPS 2022.

[2] A. Zharmagambetov, B. Amos, A. Ferber, T. Huang, B. Dilkina, and Y. Tian. Landscape Surrogate: Learning Decision Losses for Mathematical Optimization Under Partial Information. ArXiv: 2307.08964

**Questions:**

- Although this is not necessarily a weakness, but I'm just curious how this analysis will extend to other DFL methods, such as [3,4]. Could be potential direction for future research.

- What kind of specific data augmentation was performed for the first mechanism? Is it just randomly drawing A,b or some specific data augmentation techniques were applied? If yes, what was their effect?


[3] Marin Vlastelica Poganˇci ́c, Anselm Paulus, Vit Musil, Georg Martius, and Michal Rolinek. Differentiation of blackbox combinatorial solvers. In International Conference on Learning Representations, 2020.

[4] Subham S. Sahoo, Marin Vlastelica, Anselm Paulus, V ́ıt Musil, Volodymyr Kuleshov, and Georg Martius. Backpropagation through combinatorial algorithms: Identity with projection works. In International Conference on Learning Representations, 2022.

---

> ### Author Response · Authors · 2023-11-17
> **Response**
>
> **Relation to recent advancements in parametric surrogate learning**
>
> Indeed, works [1] and [2] propose ways of learning parametric surrogate losses which can then be used in a DFL loop. The authors of these works indeed characterize their methods as generalizable, but this refers to a very different kind of generalizability than the one we refer to in the present work. [1] and [2] are generalizable in the sense that they do not involve the use of hand-crafted surrogates specific to the optimization problem at hand. Instead, they only require access to a black-box solver, and thus do not exploit the structural properties of the optimization problem. In other words, they are called generalizable because they can be applied to all kinds of optimization problems. For example, they can be used to learn a surrogate for a shortest path problem, or for a knapsack problem, or for a portfolio optimization problem, etc. The techniques are generalizable in the sense that they are broadly applicable. But these works are not about how well a learned surrogate generalizes to altered optimization tasks, which is a very different kind of generalizability, and which is exactly what we refer to with ‘task generalization’ in the present work.
>
>
> **Extending the analysis to other DFL methods**
>
> There are indeed plenty of other DFL methods which differ in the way they circumvent the zero-gradient problem. In the present paper, we used SPO in our analysis since it has shown great and robust performance in existing comparative analyses (Tang & Khalil, 2022b); Mandi et al., 2023). But surely, the same analysis could be made using other techniques as well, such as the Blackbox method from [3], or the Identity method from [4]. While this would definitely be valuable to strengthen the evidence for our findings, and a good addition to a more extensive potential journal version of this article, we do not expect any different conclusions to come from the use of a different DFL technique. This is also evidenced by a similar analysis which we have performed for the learning-to-rank DFL methodologies from Mandi et al (2022), which led to the same conclusions as the analysis for SPO. Instead, our conclusions seem to relate more generally to the nature of single-task DFL: by training in a decision-focused way on a single task, the predictive model tends to overfit to that task and learns to make predictive trade-offs that work well specifically on that task, but not on other, altered tasks.
>
>
> **The kind of data augmentation**
>
> Both mechanisms (implicit and explicit generalization) sample tasks from a task distribution. The task distributions differ between the experiments. For example, in the weight generalization experiment for the knapsack problem, the different tasks have different item weights, sampled from a normal distribution (which corresponds to changes in the A matrix of the ILP formulation). The other 5 experiments all involve other task alterations. But each of these indeed corresponds to a random sampling of alterations to either A or b from some distribution, where the shape and width of this distribution differ between the experiments and between the column groups in the tables of results. For the exact specification of the task alterations for each of the 6 experiments, please see section 4.2 in the paper.
>
>
> **References**
>
> * Bo Tang and Elias Boutros Khalil. Pyepo: A pytorch-based end-to-end predict-then-optimize library with linear objective function. In OPT 2022: Optimization for Machine Learning (NeurIPS 2022 Workshop), 2022b.
> * Jayanta Mandi, James Kotary, Senne Berden, Maxime Mulamba, Victor Bucarey, Tias Guns, and Ferdinando Fioretto. Decision-focused learning: Foundations, state of the art, benchmark and future opportunities, 2023. URL https://arxiv.org/abs/2307.13565.
> * Jayanta Mandi, Vıctor Bucarey, Maxime Mulamba Ke Tchomba, and Tias Guns. Decision-focused learning: Through the lens of learning to rank. In Kamalika Chaudhuri, Stefanie Jegelka, Le Song, Csaba Szepesvari, Gang Niu, and Sivan Sabato (eds.), Proceedings of the 39th International Conference on Machine Learning, volume 162 of Proceedings of Machine Learning Research, pp. 14935–14947. PMLR, 17–23 Jul 2022. URL https://proceedings.mlr.press/v162/mandi22a.html.

---

> > ### Comment · Reviewer_jrAx · 2023-11-21
> >
> > Thank you for your response! After carefully reviewing it and other critiques, I still believe this paper makes important contributions.
> >
> > Reviewer @xQeo and others raise valid points, especially regarding experimental evaluations: it is crucial to add more realistic benchmarks in multi-task setting, which is the main focus of this paper. At the same time, given the current setting, I do believe that SPOExplGen shows significant improvement in some cases (e.g. facility location). Additionally, the empirical analysis of the current state-of-the-art DFL methods in the context of task generalization stands out as another valuable contribution of the paper.
> >
> > Regarding theoretical analysis, I believe it is hard to obtain one without resorting to unreasonable and simple assumptions.
> >
> > Considering all of this, I maintain my original score.

---

### Official Review · Reviewer_xQeo · 2023-11-01

**Soundness:** 4 excellent
**Presentation:** 4 excellent
**Contribution:** 2 fair
**Rating:** 3
**Confidence:** 4

**Summary:**

This paper studies the capability of decision-focused learning
methods to generalize to tasks beyond the training task(s).

Section 2 overviews the DFL setup where there is a dataset of
(x,c) pairs and a model m is learned to predict the linear
coefficient (c) of equation 1 from x.
This paper defines the task to be (A,b) defining the
constraints in equation 1.
The motivation for this paper is that if a model is trained on one task,
the most important parts of the c prediction may differ from other
tasks that the model is evaluated on.

Section 3 moves on to show that a model could be trained
on multiple tasks at once (in equation 5), and then suggests
that the model can be conditional (implicit, S3.1) or
unconditional (explicit, S3.2) on the task.
When conditioning on the task, the authors propose to run
a GNN on the constraints for the task to obtain a
task embedding.

Section 4 shows experimental results on knapsack,
capacitated facility location, and shortest path
problems, showing that generally the model conditional
on task information performs the best.

**Strengths:**

1. Understanding the models learned with DFL/SPO methods
   is an important research topic for the community.
2. The idea of learning task embeddings/features via a
   GNN on the constraints is interesting and novel
   as far as I am aware.
3. The experiments in Tables 1/2/3 were clearly set up
   and show the results of varying task parameters.
   Often the SPOExplGen model that explicitly conditions
   on the task performs the best. This makes sense and
   is good to experimentally demonstrate.
4. The experimental settings have the potential to become
   a benchmark for multi-task generalization in SPO methods
   (but on the other hand, are a straightforward generalization
   of existing small-scale DFL settings)

**Weaknesses:**

1. The experimental setup in section 3 is a staightforward generalization
   of existing techniques, such as Tang & Khalil (2022a).
   The beginning of section 3 states that they differ from
   Tang & Khalil (2022a) because they consider a distribution
   over tasks rather than a fixed set of tasks, but this difference
   just makes equation 5 an expectation over tasks rather than a sum.
2. Because the methodology is similar to Tang & Khalil (2022a),
   it would have been insightful to compare directly to
   a) their method on the experimental settings here,
   and/or b) the SPOImplGen and SPOExplGen methods on
   the experimental settings there.
   If possible, this could add an insightful bridge between
   the existing works, and if not possible it could be insightful
   to discuss more why not.
   (I acknowledge comparisons to their models/settings may
   not make sense if they assume more contextual information
   is available.)
3. While the experimental results through section 4 are
   scientifically well-executed and documented, I do not
   find them especially insightful or surprising.
   The problems are relatively standard for DFL and I would
   not expect methods to generalize between tasks.
   Between models, it is also unsurprising the SPOExplGen
   model conditional on the task information performs
   the best as it has the most information.

**Questions:**

I do not see any significant weaknesses in the paper
and think the paper presents an interesting scientific
investigation and experimental study of task generalization for DFL.
I lean towards rejection due to the cumulation of the smaller
weaknesses in my response. I would be especially open to re-evaluating
my score if the authors could 1) further clarify the positioning w.r.t.
related work such as Tang & Khalil (2022a), and if it's possible to compare
with their method or experiments and 2) re-emphasize any surprising
experimental results worth spreading to the community.

---

> ### Author Response · Authors · 2023-11-17
> **Response part 1**
>
> **Relation to Tang & Khalil (2022a)**
>
> The Tang & Khalil (2022a) paper addresses multi-task DFL where the set of tasks is fully known and kept the same in the training and test set. They investigate two approaches. The first is a single-cost approach, where the model outputs a single cost vector that is subsequently used in all of the tasks. This model is trained using an aggregate of the task losses for the different tasks. The second is a multi-cost approach, where the model has some shared layers, followed by a multi-headed layer, with a separate head for each task, each of which outputs a cost vector specific to the associated task. The shared layers are trained on an aggregate of the task losses for all tasks; the heads are trained specifically on the task loss for their associated task.
>
> There is a similarity between the implicit generalization training scheme and the single-cost methodology used in Tang & Khalil (2022a). Both are simple schemes methodologically, and involve training the predictive model on multiple tasks, rather than on only one. So, we argue that the single-cost contribution in Tang & Khalil (2022a) and our contribution with implicit generalization are not really of a technical nature. The contributions instead lie mostly in the insights gained, which are clearly different in the two works.
>
> The approach in Tang & Khalil (2022a) is aimed at and evaluated in terms of multi-task performance, meaning that the model is trained and evaluated on a fixed and fully-known set of tasks. That is, the evaluation is done entirely in-sample: the tasks used at test time are the same as the ones used at training time. Tang & Khalil’s main finding was that when predictions must be used in several related tasks that are known in advance, it often suffices to train one predictive model on all of these tasks together, rather than training a separate predictive model for each task.
>
> In our paper, the focus is task generalization, i.e., out-of-sample performance, on tasks that have not been seen before during training. So, rather than working with a fixed set of tasks, implicit generalization involves sampling tasks from the task distribution at training time. We found that models trained implicitly in a decision-focused way on a distribution of tasks perform much better across that distribution than models trained on individual tasks from that distribution. Additionally, we found something that runs somewhat counter to the finding in Tang & Khalil (2022a): while the models trained with implicit generalization perform well across the task distribution, they perform worse on any individual task than a model trained specifically for that task. This was the motivation for explicit task representations, which is where our work does offer a contribution that is of a more technical nature.
>
> It is somewhat unclear to us what it would mean to compare the way Tang & Khalil (2022a) train their multi-task model with the way we train our implicit generalization model. In a sense, implicit generalization is the application of their single-cost approach to the task generalization setting.
>
> Regarding the multi-cost approach of Tang & Khalil (2022a): this approach is not applicable to our experimental setting. Each head of the model is trained on a specific task, so when an out-of-sample task occurs at test time, it is unclear from which head to take the predicted cost vector. On the other hand, our training scheme with explicit representations could be applied to the experimental setting from Tang & Khalil (2022a). We chose not to do so because this lies outside of the scope of our work: our research questions relate to task generalization (i.e., out-of-sample performance), whereas the experimental setting from Tang & Khalil (2022a) is only suited to answer questions about in-sample performance. Additionally, we do not expect the explicit generalization model to generally outperform the multi-cost approach from Tang & Khalil (2022a) in their in-sample setting, since the latter has a dedicated head for each task, which is unlikely to be outperformed by a model that is trained with respect to a whole distribution of tasks (although our experiments on the facility location problem do suggest that the opposite is possible, which is a finding that must be further explored in future work).

---

> > ### Comment · Reviewer_xQeo · 2023-11-17
> >
> > Thank you for the response and clarifications on both of these! After going through the response and other reviews, I still feel that: 1) there is not much experimental significance or insights in the out-of-distribution generalization in their current form --- this seems to be shared by most other reviewers, 2) the multi-task settings come from synthetic task distributions --- it would have been more compelling to have more realistic tasks, and 3) the paper offers no new theoretical insights for generalization. For these reasons, I will maintain my original score of a reject.

---

> ### Author Response · Authors · 2023-11-17
> **Response part 2**
>
> **Summary of insights gained**
>
> We would summarize the most important new insights worth spreading to the community as follows.
>
> 1) While the application of DFL techniques generally leads to predictions with better performance on the downstream optimization task than predictions that are made to be as accurate as possible, these techniques strongly overfit the model to the optimization task at hand. This means that when the downstream task is altered at test time, the downstream task loss quickly worsens, sometimes even to the point where the decision-focused model becomes ‘counterproductive’, and where the use of a much simpler prediction-focused model leads to better downstream decisions.
>
> 2) To combat this, one may instead implicitly train on a distribution of tasks. This greatly improves the out-of-sample generalization performance, making the model more robust to task alterations that can occur after training. However, models trained implicitly on a distribution of tasks perform worse on any particular individual task from that distribution than a model trained specifically for that task, and this performance gap grows as the task distribution widens.
>
> 3) To improve on this, especially when the task alterations occur on the left-hand side of constraints, one can use our proposed explicit generalization methodology.
>
> 4) Surprisingly, on our most complex benchmark, the model trained with explicit generalization (with explicit task representations) even performed better on individual tasks from the task distribution than models trained specifically on those tasks (without explicit task representations). This means that the merit of task representations is not merely to limit the individual task performance loss incurred by training on many tasks, but that they even have the potential to improve performance on individual tasks. This finding is further discussed in the last paragraph of section 4.3, and seems to us like a great starting point for future work.

---

### Author Response · Authors · 2023-11-17
**General response**

We thank the reviewers for their valuable feedback. We are pleased to see that the importance of enabling task generalization in the context of decision-focused learning is recognized. Below, we provide a separate response to each reviewer. We also revised the paper by fixing typos and updating the references. The changes to the paper are marked in blue.

---

### Meta-Review · Area_Chair_7uLE · 2023-12-11

**Metareview:**

In this paper, the authors consider generalizing decision-focused learning beyond training tasks. The proposed method is based on the linear programming problem and exploit graph neural network for feature extraction. The authors also conducted experiments to demonstrate the benefits.

**Justification For Why Not Higher Score:**

There are several issues raised by the reviewers:

1, Although the authors provided discussion to Tang & Khalil (2022a), the discussion should be included in main text, and compared empirically.

2, Moreover, there is no explicit related work section and with several important DFL paper missing, which makes the paper not positioned well in literature.

**Justification For Why Not Lower Score:**

N/A

---

### Decision · Program_Chairs · 2024-01-16

Reject